

# Gapless chiral spin liquid from coupled chains on the kagome lattice

**Rodrigo G. Pereira[1,2]⋆ and Samuel Bieri[3,1]†**

**1** International Institute of Physics, Universidade Federal do Rio Grande do Norte, 59078-970 Natal, Brazil
**2** Departamento de Física Teórica e Experimental, Universidade Federal do Rio Grande do Norte, 59078-970 Natal, Brazil
**3** Institute for Theoretical Physics, ETH Zürich, 8099 Zürich, Switzerland

⋆ rpereira@iip.ufrn.br, † samuel.bieri@alumni.epfl.ch

## Abstract

Using a perturbative renormalization group approach, we show that the extended $(J_1 - J_2 - J_d)$ Heisenberg model on the kagome lattice with a staggered chiral interaction $(J_\chi)$ can exhibit a gapless chiral quantum spin liquid phase. Within a coupled-chain construction, this phase can be understood as a chiral sliding Luttinger liquid with algebraic decay of spin correlations along the chain directions. We calculate the low-energy properties of this gapless chiral spin liquid using the effective field theory and show that they are compatible with the predictions from parton mean-field theories with symmetry-protected line Fermi surfaces. These results may be relevant to the state observed in the kapellasite material.

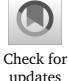

# 1   Introduction

Exotic quantum phases in frustrated magnetism have been surprising us for decades [1, 2]. A particularly interesting class, the so-called quantum spin liquid (QSL) [3–5], has recently enjoyed renewed attention due to extensive experimental advances [6, 7]. The gapped *chiral* spin liquid proposed by Kalmeyer and Laughlin [8] was an early theoretical example, exhibiting properties such as anyon excitations, previously only predicted in the fractional quantum Hall effect [9–13].

Chiral spin liquids have recently been a particularly active field of research, both theoretically and in materials science [14–23]. Partially, this is due to our deeper understanding of new concepts such as symmetry fractionalization and the classification of gapped topological phases [24–28]. Another important milestone, however, was the characterization of materials with further-neighbor exchange interactions, such as the QSL candidate kapellasite [29–33]. As it turns out, such interactions can spontaneously induce chiral phases [34–38]. Interestingly, none of the currently available candidate materials show clear signatures of a spin gap. It appears that spin liquids with *gapless* spin excitations may be more common in nature than topological phases.

A highly fruitful approach to gain insight into the phenomenology of gapless QSLs has been the so-called parton construction, where spin is fractionalized into fermionic spinon operators [39–45]. For some integrable spin models, it is even possible to construct exact eigenstates in this way [46–50]. However, in general the parton construction entails a subtle mean-field approximation that relaxes local constraints and neglects fluctuations of the emergent gauge field. In particular, it remains debated whether U(1) QSLs with a spinon Fermi surface can be stable against gauge fluctuations [4].

An alternative route to constructing QSLs in two dimensions employs antiferromagnetic spin chains as elementary building blocks. Previous works in this direction have mainly concentrated on gapped QSLs [51–56] and can be viewed as variants of coupled-wire constructions of topological phases [57–61]. A natural question is whether one can construct a gapless liquid phase using this approach. Starykh *et al.* [62] proposed that a model of crossed spin-1/2 Heisenberg chains might stabilize a "sliding Luttinger liquid", in which interchain couplings are irrelevant in the renormalization group (RG) sense. At zero temperature, such a phase would have power-law decaying correlations along the chain directions. The proposal was later dismissed [63] with the argument that, even in a highly frustrated spin system, there are always residual interactions which are relevant and eventually drive the system to either

a dimerized or a classical, magnetically ordered phase.

In this paper, we show that a gapless QSL constructed from weakly coupled spin chains may be stabilized when time-reversal symmetry is explicitly broken and one of the two chiral modes on every chain is gapped out. We start from a model of crossed spin chains formed by a dominant antiferromagnetic exchange interaction $J_d$ across the hexagons of the kagome lattice. Such a model has recently been investigated in relation with kapellasite [36, 37]. Using perturbative RG, we show that time-reversal-breaking interchain couplings generated by scalar chirality terms can flow to strong coupling before all other competing interactions. We conjecture that this flow to strong coupling signals a full gap for the collective spin modes in one chiral sector, while the other chiral sector remains gapless. We show that the corresponding theory is a stable fixed point in the RG. The resulting two-dimensional state is an exotic chiral spin liquid with persistent bulk spin currents, gapless excitation continua, algebraic decay of correlation functions, and logarithmic violation of the entanglement area law. Since these properties are usually associated with fermionic spinons, we also pursue the goal of identifying the parton construction that describes this phase. We find that the properties predicted within the coupled-chains construction are consistent with fermionic partons that exhibit line Fermi surfaces protected by reflection symmetry.

The paper is organized as follows. In Section 2, we introduce the Heisenberg spin model on the kagome lattice that we want to study. In Section 3, we discuss the continuum theory of decoupled crossed chains at dominant $J_d$ and we introduce the fields and their operator product expansion. In Section 4, we derive the effective two-dimensional field theory and we investigate the effect of the chiral perturbation $J_\chi$ in the compensated regime $J_1 = J_2$. In Section 5, we solve the renormalization group equations and we present the phase diagram for the model. For sufficiently strong chiral interaction, we find a novel gapless chiral spin liquid, and we discuss its physical properties in Section 6. In Section 7, we propose parton mean-field theories with lines of spinon Fermi surface that can reproduce the intriguing physical properties of the found quantum spin liquid. Finally, we conclude in Section 8.

## 2 Lattice model

We want to analyze a Heisenberg model on the kagome lattice with first-neighbor $J_1$, second-neighbor $J_2$, and exchange interaction $J_d$ across the diagonals of the hexagons. In the case of dominant $J_d \gg |J_1|, |J_2|$, the model effectively decouples into three arrays of chains running at 120° [36]. In the following, we adopt the notations of Ref. [37]. The chain directions are labelled by $q = 1, 2, 3 \in \mathbb{Z}_3$ in the cyclic group. Each site belongs to a single chain, and the spin operators on $q$-chains are denoted by $\mathbf{S}_q(j, l)$, with $j, l \in \mathbb{Z}$. The first argument gives the position along the chain, while parallel chains are indexed by the second argument. This labelling convention is illustrated in Fig. 1.

In the limit of decoupled chains ($J_1 = J_2 = 0$), the model is

$$H_0 = J_d \sum_{q,j,l} \mathbf{S}_q(j,l) \cdot \mathbf{S}_q(j+1,l). \tag{1}$$

Exchange interactions between first and second neighbors couple chains running in different directions. This can be written as [37]

$$H' = J_1 \sum_{q,j,l} \{\mathbf{S}_q(-j,l) \cdot \mathbf{S}_{q+1}(j+l-1,j) + \mathbf{S}_q(-j-1,l) \cdot \mathbf{S}_{q+1}(j+l,j)\}$$

$$+ J_2 \sum_{q,j,l} \{\mathbf{S}_q(-j-1,l) \cdot \mathbf{S}_{q+1}(j+l-1,j) + \mathbf{S}_q(-j,l) \cdot \mathbf{S}_{q+1}(j+l,j)\}. \tag{2}$$

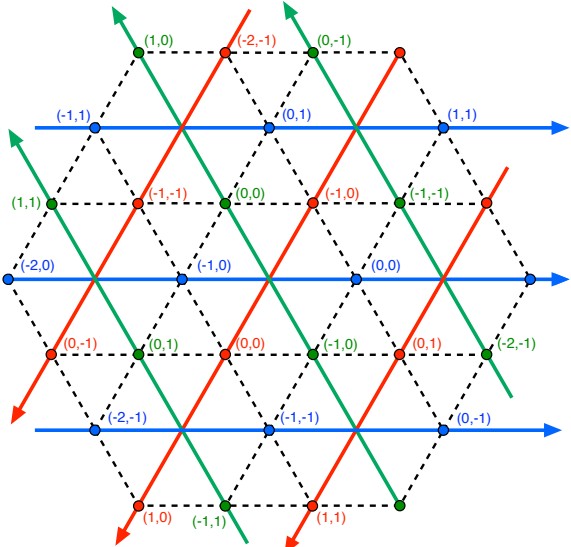

Figure 1: Crossed-chain model on the kagome lattice. Solid lines represent the bonds with dominant exchange coupling $J_d$. There are three arrays of parallel chains depicted in different colors (blue, green, and red for $q = 1, 2, 3$, respectively), and the sites are labelled by $(j, l)$ in Eq. (1). The arrows indicate the positive-$x$ direction within each chain, which becomes the direction of propagation of right-moving bosonic modes in the continuum limit. Dashed lines represent first-neighbor bonds of the lattice.

Recently, it has been shown within perturbative and numerical density-matrix RG [37] that $H'$ induces nonplanar magnetic orderings of "cuboc" [34] type with a 12-site cell for sufficiently large $|J_1 - J_2|$. In the *compensated regime* $J_1 \simeq J_2$, these authors found that a valence bond crystal with a 24-fold degenerate ground state is stabilized.[1] In the QSL candidate material kapellasite, these couplings are found to be weakly ferromagnetic ($J_1, J_2 < 0$) [29–31].

To probe for novel phases, we add scalar chirality terms $\mathbf{S}_i \cdot (\mathbf{S}_j \times \mathbf{S}_k)$ to the model, explicitly breaking time reversal and some point group symmetries of the lattice. We choose the staggered chirality pattern illustrated in Fig. 2. For each hexagon, we turn on chiral interactions on triangles formed by one first neighbor, one second neighbor, and one bond across the hexagon diagonal. In each of these triangles, two sites belong to the same chain and the third site belongs to a different chain. The chiral interaction is staggered in a way that preserves all reflections $\sigma_q$ along $q$-chains, but breaks reflection symmetry $\sigma'_q$ on lines *perpendicular* to $q$-chains. Since the $\pi/3$-rotation $R$ can be written as $R = \sigma'_q \sigma_{q-1}$, it follows that $R$ is necessarily broken [45]. More explicitly, our chiral interaction corresponds to the term

$$H_\chi = J_\chi \sum_{q,j,l} [\mathbf{S}_q(-j-1, l) \times \mathbf{S}_q(-j, l)] \cdot [\mathbf{S}_{q+1}(j+l, j) + \mathbf{S}_{q+1}(j+l-1, j)$$

$$+ \mathbf{S}_{q+2}(-l-1, -j-l) + \mathbf{S}_{q+2}(-l, -j-l)]. \quad (3)$$

This particular choice of chirality pattern will be justified and motivated by the analysis of the interactions generated in the effective field theory discussed in Section 4.

Let us discuss the symmetries of the lattice model in more detail. Translation symmetry

---

[1]Earlier variational Monte Carlo investigations of this model found gapless quantum spin liquids in the parameter range $|J_1|, |J_2| \ll J_d$ [36]. That conclusion may be biased due to the great difficulty in constructing good variational wave functions with nonplanar Néel or valence-bond-crystal orders.

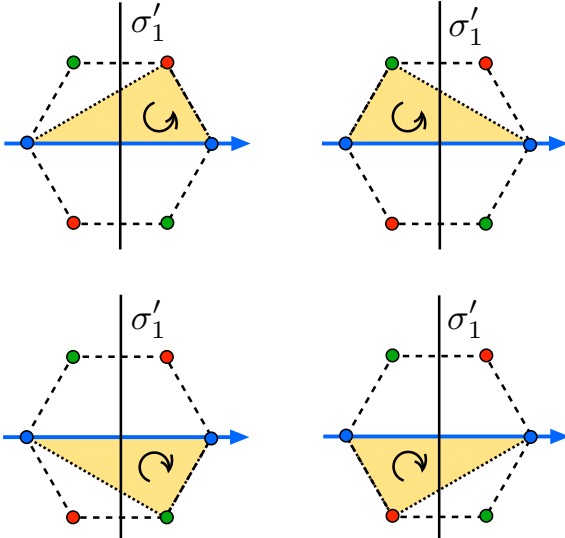

Figure 2: Scalar spin chirality terms in Eq. (3) that involve a pair of spins on the same chain (here, with $q = 1$ direction). These chiralities are staggered under $\pi/3$ lattice rotation $R$. This pattern breaks the symmetry under mirror reflection $\sigma'_1$ perpendicular to the chain, but it preserves reflection $\sigma_1$ along the chain.

along the chain direction $q$ can be written as

$$T_q : \begin{cases} (j,l)_q & \mapsto & (j+1,l)_q \\ (j,l)_{q+1} & \mapsto & (j-1,l-1)_{q+1} \\ (j,l)_{q+2} & \mapsto & (j,l+1)_{q+2} \end{cases}. \tag{4}$$

For $J_\chi \neq 0$, the point group is generated by the $2\pi/3$ rotation $R^2$ and the reflections $\sigma_q$. The action of $R^2$ performs a cyclic permutation of the chain directions,

$$R^2 : (j,l)_q \mapsto (j,l)_{q+1}. \tag{5}$$

The reflection symmetry is

$$\sigma_q : (j,l)_{q+p} \mapsto (j-l,-l)_{q-p}. \tag{6}$$

Note the additional translation in the chain direction.

The $\pi/3$ rotation is

$$R : (j,l)_q \mapsto (-j-1,-l)_{q+2}. \tag{7}$$

This symmetry is broken for $J_\chi \neq 0$. However, its composition with time reversal

$$\Theta : \mathbf{S} \mapsto -\mathbf{S} \tag{8}$$

yields the symmetry $\Theta R$. Similarly, the reflection $\sigma'_q = R\sigma_{q-1}$ (across a line perpendicular to a $q$-chain),

$$\sigma'_q : (j,l)_{q+p} \mapsto (l-j-1,l)_{q-p}, \tag{9}$$

is broken but $\Theta\sigma'_q$ is preserved in our model. Note that $\sigma'_q$ alternates between bond and site reflection for $q$-chains, depending on the parity of $l$ (see Fig. 1). This fact will be important in the continuum limit.

# 3 Continuum limit

We want to derive an effective field theory for our lattice model, with the goal to analyze the effect of the interchain interactions in Eqs. (2) and (3) on the decoupled-chain Hamiltonian Eq. (1). In a first step, we introduce the continuum limit of the decoupled chains and the algebra of field operators. Next, we discuss how the two-dimensional limit can be taken in order to incorporate the interchain couplings.

## 3.1 Decoupled chains

For $J_d > 0$ and $J_1 = J_2 = J_\chi = 0$, we can bosonize the spin operators in each chain and write [64]

$$\mathbf{S}_q(j,l) \sim a_\parallel \mathbf{m}_q(x,l) + (-1)^j A \sqrt{a_\parallel}\, \mathbf{n}_q(x,l), \tag{10}$$

where $a_\parallel = 2a$ with $a$ the (nearest-neighbor) kagome lattice spacing and and $A$ is a dimensionless nonuniversal constant of order one. The position along the chain is $j$, and $a_\parallel j \to x$ in the continuum limit. The fields $\mathbf{m}_q(x,l)$ and $\mathbf{n}_q(x,l)$ are smooth functions of their first argument. The uniform part is the sum of the chiral currents of the SU(2)$_1$ Wess-Zumino-Witten (WZW) model,

$$\mathbf{m}_q(x,l) = \mathbf{J}_{qL}(x,l) + \mathbf{J}_{qR}(x,l). \tag{11}$$

Using abelian bosonization, we write the components of the chiral currents in the form

$$J^z_{q\nu}(x,l) = \nu \partial_x \varphi_{q\nu}(x,l)/\sqrt{4\pi}, \tag{12a}$$

$$J^\pm_{q\nu}(x,l) = e^{\pm i\sqrt{4\pi}\varphi_{q\nu}(x,l)}/(2\pi), \tag{12b}$$

where $\nu = L/R = +/-$, and the chiral bosonic fields obey

$$[\varphi_{q\nu}(x,l), \partial_x \varphi_{q'\nu'}(x',l')] = i\nu \delta_{qq'}\delta_{\nu\nu'}\delta_{ll'}\delta(x-x'). \tag{13}$$

Here we use the "CFT normalization condition" [65] for the vertex operators. The components of the staggered magnetization can be written as

$$n^z_q(x,l) = \sin[\sqrt{2\pi}\phi_q(x,l)], \tag{14a}$$

$$n^\pm_q(x,l) = \pm i e^{\pm i\sqrt{2\pi}\theta_q(x,l)}, \tag{14b}$$

where $\phi_q = (\varphi_{qL} - \varphi_{qR})/\sqrt{2}$ and $\theta_q = (\varphi_{qL} + \varphi_{qR})/\sqrt{2}$. In addition to the operators appearing in Eq. (10), we also need the dimerization operator

$$\varepsilon_q(x,l) = \cos[\sqrt{2\pi}\phi_q(x,l)]. \tag{15}$$

The operators $\mathbf{J}_{q\nu}$, $\mathbf{n}_q$, and $\varepsilon_q$ form a closed operator algebra. The list of operator product expansions (OPEs) is analogous to the one given by Metavitsiadis *et al.* [66].[2]

The continuum limit of the Hamiltonian for decoupled chains can now be written as

$$H_0 \sim \sum_{q,l} \frac{2\pi v}{3} \int dx \, [\mathbf{J}^2_{qL}(x,l) + \mathbf{J}^2_{qR}(x,l)] + 2\pi v \gamma_{\mathrm{bs}} \int dx \, \mathbf{J}_{qL}(x,l) \cdot \mathbf{J}_{qR}(x,l), \tag{16}$$

where $v = \pi J_d a$ is the spin velocity and $\gamma_{\mathrm{bs}} \sim \mathcal{O}(1)$ is the dimensionless coupling constant of the marginal "backscattering" operator. The latter is marginally irrelevant for $\gamma_{\mathrm{bs}} < 0$ and accounts for logarithmic corrections to low-energy properties of the antiferromagnetic Heisenberg chain [67]. Dropping the $\gamma_{\mathrm{bs}}$-term, the Hamiltonian in Eq. (16) is equivalent to a set of

---

[2]We choose $\gamma = 1/(2\pi)$ in the notation of these authors [66].

free chiral bosons $\varphi_{q\nu}(x,l)$, which enables us to calculate correlation functions by standard methods [68]. In particular, the OPE for the chiral currents is

$$J^a_{qL}(x,l,\tau)J^b_{q'L}(0,l',0) \sim \delta_{qq'}\delta_{ll'}\left[\frac{\delta^{ab}}{8\pi^2 z^2} + \frac{i\epsilon^{abc}}{2\pi z}J^c_{qL}(0,l,0)\right], \qquad (17a)$$

$$J^a_{qR}(x,l,\tau)J^b_{q'R}(0,l',0) \sim \delta_{qq'}\delta_{ll'}\left[\frac{\delta^{ab}}{8\pi^2 \bar{z}^2} + \frac{i\epsilon^{abc}}{2\pi \bar{z}}J^c_{qR}(0,l,0)\right], \qquad (17b)$$

where the operators evolve in imaginary time $\tau$ and we define $z = v\tau + ix$, $\bar{z} = v\tau - ix$. Note that, since the chains are not parallel, the direction of motion of the bosonic modes in the two-dimensional plane depends on $q$: for each $q$, the $R$ modes are the ones that move in the positive-$x$ direction indicated by the arrows in Fig. 1.

### 3.2 Two-dimensional theory

Once we turn on interchain couplings, it becomes important to take the continuum limit in the second argument of $\mathbf{J}_{q\nu}(x,l)$ as well. The reason is that the interactions in Eqs. (2) and (3) mix the coordinates $(j,l)$ of chains with different $q$. In practice, we are after an effective theory in $2+1$ dimensions that describes a set of spin chains which are "coarse grained" in the perpendicular direction. The procedure amounts to restricting the states in momentum space to those with $Q_\perp a \ll 1$, where $Q_\perp$ is the momentum perpendicular to the chains [69].

To take the continuum limit in the perpendicular direction, we first note that, from a symmetry perspective, subsequent parallel chains cannot be treated in a completely identical way. As discussed, a reflection $\sigma'_q$ acts on every other $q$-chain as a bond reflection, while it acts as a site reflection on the remaining perpendicular chains (see Figs. 1 and 2). We thus take into account a potential staggering in the transverse chain direction by doubling the number of fields. We define $\mathbf{J}_{q\nu e}$ and $\mathbf{J}_{q\nu o}$ such that

$$\mathbf{J}_{q\nu}(x,l) \sim \sqrt{a_\perp}\left[\mathbf{J}_{q\nu e}(x,y) + (-1)^l \mathbf{J}_{q\nu o}(x,y)\right], \qquad (18)$$

where $a_\perp = 2\sqrt{3}a$ and we take $a_\perp l \to y$ in the continuum limit. The inverse transformation is (for $y/a_\perp = 2\ell$; i.e. $\ell = \lfloor l/2 \rfloor$)

$$\mathbf{J}_{q\nu e}(x,y) \sim \frac{1}{2\sqrt{a_\perp}}\left[\mathbf{J}_{q\nu}(x,2\ell) + \mathbf{J}_{q\nu}(x,2\ell-1)\right], \qquad (19a)$$

$$\mathbf{J}_{q\nu o}(x,y) \sim \frac{1}{2\sqrt{a_\perp}}\left[\mathbf{J}_{q\nu}(x,2\ell) - \mathbf{J}_{q\nu}(x,2\ell-1)\right]. \qquad (19b)$$

Thus, the indices $e/o$ label fields which are even/odd under permutation of even and odd parallel chains (e.g., by lattice translation, $2\ell \mapsto 2\ell-1$). In terms of these new fields, Eq. (16) can be rewritten in the form

$$H_0 \sim 2\pi v \sum_q \int dx\, dy \left\{\frac{1}{3}\sum_\nu\left[\mathbf{J}^2_{q\nu e}(x,y) + \mathbf{J}^2_{q\nu o}(x,y)\right]\right.$$

$$\left. + \gamma_{\mathrm{bs}}\left[\mathbf{J}_{qLe}(x,y)\cdot\mathbf{J}_{qRe}(x,y) + \mathbf{J}_{qLo}(x,y)\cdot\mathbf{J}_{qRo}(x,y)\right]\right\}. \qquad (20)$$

Similarly, we define $e/o$ fields for the staggered magnetization and for the dimerization operators such that

$$\mathbf{n}_q(x,l) \sim \sqrt{a_\perp}\left[\mathbf{n}_{qe}(x,y) + (-1)^l \mathbf{n}_{qo}(x,y)\right], \qquad (21a)$$

$$\varepsilon_q(x,l) \sim \sqrt{a_\perp}\left[\varepsilon_{qe}(x,y) + (-1)^l \varepsilon_{qo}(x,y)\right]. \qquad (21b)$$

In terms of these fields, the mode expansion for the spin operator in Eq. (10) becomes

$$
\mathbf{S}_q(j,l) \sim a_\parallel \sqrt{a_\perp} \big[ \mathbf{m}_{qe}(x,y) + (-1)^l \mathbf{m}_{qo}(x,y)
$$
$$
+ (-1)^j A a_\parallel^{-1/2} \mathbf{n}_{qe}(x,y) + (-1)^{j+l} A a_\parallel^{-1/2} \mathbf{n}_{qo}(x,y) \big]. \tag{22}
$$

The OPEs for $e/o$ fields can be obtained from Eqs. (19). For instance, we have

$$
J_{qLe}^a(x,y,\tau) J_{q'Le}^b(0,y',0) \sim \frac{\delta_{qq'}}{2} \left[ \frac{\delta_{\ell\ell'}}{a_\perp} \frac{\delta^{ab}}{8\pi^2 z^2} + \frac{\delta_{\ell\ell'}}{\sqrt{a_\perp}} \frac{i\epsilon^{abc}}{2\pi z} J_{qLe}^c(0,y,0) \right]. \tag{23}
$$

At this point, we have to decide how to handle the factors of $\delta_{\ell\ell'}$ in the continuum limit. These factors tell us that two parallel chains are not correlated when the distance $|y - y'|$ is larger than the short-distance cutoff $a_\perp$. We introduce the regularized delta function

$$
\Delta(x,\alpha) = \alpha^{-1} e^{-\pi x^2/\alpha^2}, \tag{24}
$$

such that $\Delta(0,\alpha) = \alpha^{-1}$ and $\int_{-\infty}^{+\infty} dx\, \Delta(x,\alpha) = 1$. We rewrite Eq. (23) as

$$
J_{qLe}^a(x,y,\tau) J_{q'Le}^b(0,y',0) \sim \frac{\delta_{qq'}}{2} \bigg\{ \Delta(y-y',a_\perp) \frac{\delta^{ab}}{8\pi^2 z^2}
$$
$$
+ \sqrt{\Delta(y-y',a_\perp)} \frac{i\epsilon^{abc}}{2\pi z} J_{qLe}^c(0,y,0) \bigg\}. \tag{25}
$$

With this prescription, all factors of short-distance cutoff are absorbed in the regularized delta function. In the same way, we obtain

$$
J_{qLo}^a(x,y,\tau) J_{q'Lo}^b(0,y',0) \sim \frac{\delta_{qq'}}{2} \bigg\{ \Delta(y-y',a_\perp) \frac{\delta^{ab}}{8\pi^2 z^2}
$$
$$
+ \sqrt{\Delta(y-y',a_\perp)} \frac{i\epsilon^{abc}}{2\pi z} J_{qLe}^c(0,y,0) \bigg\} \tag{26}
$$

and

$$
J_{qLe}^a(x,y,\tau) J_{q'Lo}^b(0,y',0) \sim \frac{\delta_{qq'}}{2} \sqrt{\Delta(y-y',a_\perp)} \frac{i\epsilon^{abc}}{2\pi z} J_{qLo}^c(0,y,0). \tag{27}
$$

Other OPEs that will be used in the following are

$$
J_{qLo}^a(x,y,\tau) \partial_x n_{qe}^b(0,y',0) \sim \frac{\sqrt{\Delta(y-y',a_\perp)}}{8\pi z^2} \big\{ \delta^{ab} \varepsilon_{qo}(0,y,0) - \epsilon^{abc} n_{qo}^c(0,y,0) \big\}, \tag{28a}
$$

$$
J_{qRo}^a(x,y,\tau) \partial_x n_{qe}^b(0,y',0) \sim \frac{\sqrt{\Delta(y-y',a_\perp)}}{8\pi z^2} \big\{ \delta^{ab} \varepsilon_{qo}(0,y,0) + \epsilon^{abc} n_{qo}^c(0,y,0) \big\}. \tag{28b}
$$

Note the fusion rules $e \times e \to e$, $o \times o \to e$ and $e \times o \to o$ in Eqs. (25)–(28).

In the continuum limit, the symmetries discussed in Section 2 must be supplemented by transformation rules for the fields. Let us first recall the symmetries of a single chain. Under time reversal $\Theta$, we have

$$
\mathbf{J}_\nu \mapsto -\mathbf{J}_{-\nu}, \quad \mathbf{n} \mapsto -\mathbf{n}, \quad \varepsilon \mapsto \varepsilon. \tag{29}
$$

Translation along the chain is

$$
\mathbf{J}_\nu \mapsto \mathbf{J}_\nu, \quad \mathbf{n} \mapsto -\mathbf{n}, \quad \varepsilon \mapsto -\varepsilon, \tag{30}
$$

while site inversion is given by

$$
\mathbf{J}_\nu \mapsto \mathbf{J}_{-\nu}, \quad \mathbf{n} \mapsto \mathbf{n}, \quad \varepsilon \mapsto -\varepsilon, \tag{31}
$$

and bond inversion is

$$\mathbf{J}_\nu \mapsto \mathbf{J}_{-\nu}, \quad \mathbf{n} \mapsto -\mathbf{n}, \quad \varepsilon \mapsto \varepsilon. \tag{32}$$

Furthermore, spin rotation symmetry $\mathscr{S}$ is implemented as global SO(3) rotation of $\mathbf{J}_\nu$ and $\mathbf{n}$.

Progressing to the two-dimensional crossed-chain model, we must examine the effect of the kagome-lattice symmetries on the $e/o$ fields. The important new rule stems from

$$\sigma_q : \begin{cases} \mathbf{n}_q(x,l) & \mapsto (-1)^l \, \mathbf{n}_q(x-l,-l) \\ \varepsilon_q(x,l) & \mapsto (-1)^l \, \varepsilon_q(x-l,-l) \end{cases}. \tag{33}$$

Since we can write

$$\mathbf{n}_{q,e/o}(x,y) \sim \frac{1}{2\sqrt{a_\perp}} \Big[ \mathbf{n}_q(x,2\ell) \pm \mathbf{n}_q(x,2\ell-1) \Big], \tag{34a}$$

$$\varepsilon_{q,e/o}(x,y) \sim \frac{1}{2\sqrt{a_\perp}} \Big[ \varepsilon_q(x,2\ell) \pm \varepsilon_q(x,2\ell-1) \Big], \tag{34b}$$

we conclude that reflection $\sigma_q$ exchanges $e$ and $o$ for the operators $\varepsilon$ and $\mathbf{n}$,

$$\sigma_q : \begin{cases} \mathbf{n}_{q+p,e} & \mapsto \mathbf{n}_{q-p,o} \\ \mathbf{n}_{q+p,o} & \mapsto \mathbf{n}_{q-p,e} \\ \varepsilon_{q+p,e} & \mapsto \varepsilon_{q-p,o} \\ \varepsilon_{q+p,o} & \mapsto \varepsilon_{q-p,e} \end{cases}, \tag{35}$$

whereas the indices $e/o$ in the chiral currents are invariant under $\sigma_q$. On the other hand, one can easily show that the reflection $\sigma_q'$ (perpendicular to chains) does not exchange $e$ and $o$ for any field. However, left- and right-moving currents are permuted as is manifest from Eqs. (31) and (32).

## 4 Interchain couplings in the effective field theory

Let us now discuss the perturbations to the decoupled-chains Hamiltonian Eq. (1). We start with the terms generated by $H'$ in Eq. (2). We will focus on the compensated regime $J_1 = J_2$, where the model has a high degree of geometric frustration and stable nonclassical phases are most plausible. In this case, we have

$$H' = J_1 \sum_{q,j,l} \Big[ \mathbf{S}_q(-j,l) + \mathbf{S}_q(-j-1,l) \Big] \cdot \Big[ \mathbf{S}_{q+1}(j+l,j) + \mathbf{S}_{q+1}(j+l-1,j) \Big]. \tag{36}$$

Using the mode expansion in Eq. (22) and dropping rapidly oscillating terms, we obtain

$$H' \sim 2J_1 a_\parallel \sum_q \int dx\, dy \Big[ 2\, \mathbf{m}_{qe}(-x,y) \cdot \mathbf{m}_{q+1,e}(x+y,x)$$

$$+ A\sqrt{a_\parallel}\, \mathbf{m}_{qo}(-x,y) \cdot \big\{ \partial_1 \mathbf{n}_{q+1,o}(x+y,x) + \partial_1 \mathbf{n}_{q-1,e}(x+y,x) \big\} \Big], \tag{37}$$

where $\partial_1$ denotes the derivative with respect to the first argument. Equation (37) is manifestly invariant under all model symmetries discussed in the last section. The first term in this equation involves chiral currents only; we will show later that such a term behaves as a marginal operator. The last two terms represent irrelevant couplings. As shown by Gong et al. [37], the leading relevant operator that couples the staggered magnetization in different chains is $\mathbf{n}_{qo}(-x,y) \cdot \mathbf{n}_{q+1,e}(x+y,x)$. This term induces magnetic ordering, but it comes with a prefactor $(J_1 - J_2)$ in $H'$ that cancels in the compensated regime considered here.

Next, consider the perturbation $H_\chi$ in Eq. (3). This term involves the cross product between two nearest-neighbor spins of a chain, and, in the continuum limit, we have [52]

$$\mathbf{S}_q(-j-1,l) \times \mathbf{S}_q(-j,l) \sim \frac{a_\parallel}{\pi}(1+A^2)\big[\mathbf{J}_{qR}(-x,l)-\mathbf{J}_{qL}(-x,l)\big]. \tag{38}$$

This can be recognized as the density of spin current. Hereafter, we set the value of the nonuniversal constant $A = 1$. Using the mode expansion in Eq. (22), we find

$$H_\chi \sim \frac{4J_\chi a_\parallel}{\pi} \sum_q \int dx\,dy \Big[ 2\,\mathbf{K}_{qe}(-x,y) \cdot \mathbf{m}_{q+1,e}(x+y,x)$$
$$+ \sqrt{a_\parallel}\,\mathbf{K}_{qo}(-x,y) \cdot \big\{ \partial_1\mathbf{n}_{q+1,o}(x+y,x) + \partial_1\mathbf{n}_{q-1,e}(x+y,x) \big\} \Big], \tag{39}$$

where

$$\mathbf{K}_{q,e/o}(x,y) = \mathbf{J}_{qR,e/o}(x,y) - \mathbf{J}_{qL,e/o}(x,y) \tag{40}$$

is the density of $e/o$ spin currents. Note that $\mathbf{K}_{q,e/o}$ is invariant while the other terms in Eq. (39) are odd under time reversal. Conversely, $\mathbf{K}_{q,e/o}$ changes sign under reflections $\sigma'_p$ while the other terms are invariant. Similar to Eq. (37), we expect the first term in Eq. (39) to behave as a marginal operator. The other terms in Eq. (39) are irrelevant.

The irrelevant operators in Eqs. (37) and (39) cannot be neglected at the outset of the RG flow because they generate relevant operators to second order in perturbation theory. To see this, consider the expansion of

$$\hat{T} = T_\tau e^{-\int d\tau\,[H'(\tau)+H_\chi(\tau)]}, \tag{41}$$

where $T_\tau$ stands for imaginary-time ordering. To second order in $J_1$, we obtain

$$\hat{T} \sim 4J_1^2 a_\parallel^3 \sum_q \int d^3x_1\,d^3x_2\,m_{qo}^a(-x_1,y_1,\tau_1)\,\partial_1 n_{qe}^b(-x_2,y_2,\tau_2)$$
$$\times \partial_1 n_{q+1,o}^a(x_1+y_1,x_1,\tau_1)\,m_{q+1,o}^b(x_2+y_2,x_2,\tau_2). \tag{42}$$

We can now use the OPEs and integrate out short spacetime separations to find the relevant operators. Since the chains are uncorrelated at the initial steps of the RG, we use a regularized delta function with $\alpha_0 \ll a_\perp$. For concreteness, we take $\alpha_0 = a_\perp/10$. We then integrate out arbitrarily large distances, $-\infty < x = x_1 - x_2 < \infty$ and $-\infty < y = y_1 - y_2 < \infty$, but only short times $a_\perp/(2v) < |\tau| = |\tau_1 - \tau_2| < a_\perp/v$ in Eq. (42). The precise choice of the spatial cutoff and the short-time interval affect the nonuniversal value of the bare coupling constants. To second order in $J_1$ and $J_\chi$, and using the OPEs (25)–(28), this procedure generates the following terms in the Hamiltonian:

$$H_\varepsilon = \kappa_\varepsilon \sum_q \int dx\,dy\,\varepsilon_{qo}(-x,y)\,\varepsilon_{q+1,e}(x+y,x), \tag{43a}$$

$$H_n = \kappa_n \sum_q \int dx\,dy\,\mathbf{n}_{qo}(-x,y) \cdot \mathbf{n}_{q+1,e}(x+y,x). \tag{43b}$$

The bare couplings are

$$\kappa_\varepsilon = -\frac{3a_\parallel^3}{4\pi v a_\perp^2}\left( C_1 J_1^2 + \frac{4}{\pi^2} C_2 J_\chi^2 \right), \tag{44a}$$

$$\kappa_n = \frac{a_\parallel^3}{2\pi v a_\perp^2}\left( C_2 J_1^2 + \frac{4}{\pi^2} C_1 J_\chi^2 \right), \tag{44b}$$

where $C_{1,2} > 0$ are constants given in Appendix A. The dimerization coupling $\kappa_\varepsilon < 0$ favors a valence-bond-crystal order. The coupling $\kappa_n > 0$ (involving the staggered magnetization) drives the system to the *cuboc-1* phase [37].

Putting together the results in Eqs. (37), (39), and (43), we find that the list of leading perturbations to the free-boson model in Eq. (16) can be organized as follows:

$$\delta H = \sum_q \int dx\, dy \sum_{i=1}^{8} \mathscr{H}_i(x,y), \tag{45}$$

with

$$\mathscr{H}_1(x,y) = 2\pi v \lambda_1 \alpha^{-1} \varepsilon_{qo}(-x,y)\,\varepsilon_{q+1,e}(x+y,x), \tag{46a}$$

$$\mathscr{H}_2(x,y) = 2\pi v \lambda_2 \alpha^{-1} \mathbf{n}_{qo}(-x,y)\cdot\mathbf{n}_{q+1,e}(x+y,x), \tag{46b}$$

$$\mathscr{H}_3(x,y) = 2\pi v \lambda_3 \mathbf{J}_{qLe}(x,y)\cdot\mathbf{J}_{qRe}(x,y), \tag{46c}$$

$$\mathscr{H}_4(x,y) = 2\pi v \lambda_4 \mathbf{J}_{qLo}(x,y)\cdot\mathbf{J}_{qRo}(x,y), \tag{46d}$$

$$\mathscr{H}_5(x,y) = 2\pi v \lambda_5 \mathbf{J}_{qLe}(-x,y)\cdot\mathbf{J}_{q+1,Le}(x+y,x), \tag{46e}$$

$$\mathscr{H}_6(x,y) = 2\pi v \lambda_6 \mathbf{J}_{qRe}(-x,y)\cdot\mathbf{J}_{q+1,Re}(x+y,x), \tag{46f}$$

$$\mathscr{H}_7(x,y) = 2\pi v \lambda_7 \mathbf{J}_{qLe}(-x,y)\cdot\mathbf{J}_{q+1,Re}(x+y,x), \tag{46g}$$

$$\mathscr{H}_8(x,y) = 2\pi v \lambda_8 \mathbf{J}_{qRe}(-x,y)\cdot\mathbf{J}_{q+1,Le}(x+y,x), \tag{46h}$$

where $\alpha$ is the short-distance cutoff and $\lambda_i$ are dimensionless coupling constants. The bare values derived from above analysis are

$$\lambda_1^0 = \frac{\kappa_\varepsilon \alpha_0}{2\pi v}, \tag{47a}$$

$$\lambda_2^0 = \frac{\kappa_n \alpha_0}{2\pi v}, \tag{47b}$$

$$\lambda_3^0 = \lambda_4^0 = \gamma_{\text{bs}}, \tag{47c}$$

$$\lambda_5^0 = \lambda_7^0 = \frac{2a_\parallel}{\pi v}\left(J_1 - \frac{2J_\chi}{\pi}\right), \tag{47d}$$

$$\lambda_6^0 = \lambda_8^0 = \frac{2a_\parallel}{\pi v}\left(J_1 + \frac{2J_\chi}{\pi}\right). \tag{47e}$$

In Section 5, we will show that $\lambda_1$ and $\lambda_2$ are relevant perturbations, while $\lambda_3$ through $\lambda_6$ are marginal. Equations (46) contain all perturbations allowed by the lattice and spin-rotation symmetries of the model, $\{T_q, \sigma_q, \Theta\sigma'_q, \mathscr{S}\}$ (up to irrelevant operators that contain derivatives of fields). Breaking of time reversal is manifest in the marginal operators when $\lambda_5 \neq \lambda_6$ and $\lambda_7 \neq \lambda_8$. Remarkably, for strong enough chirality parameter ($|J_\chi| > \pi|J_1|/2$), the coupling $\lambda_5$ between left-moving currents starts off with a different sign than the coupling $\lambda_6$ between right-moving currents. This leads to an interesting regime in which the interaction can flow to strong coupling in one chiral sector, while it flows to weak coupling in the other sector.

At this point, we can further elucidate the criterion for fixing the chirality pattern discussed in Section 2: The contributions from the four triangles shown in Fig. 2 should add up in such a way that the three-spin interaction generates time-reversal-odd $\mathbf{J}_q \cdot \mathbf{J}_{q+1}$ couplings to first order in $J_\chi$. Intuitively, the chirality on these four triangles is such that the arrows always point in the positive $x$ direction of the reference ($q = 1$) chain. This suggests that the chiral interaction favors strong coupling between right-moving spin currents in each chain, while left movers may remain free. In the next section, we will show that this interaction asymmetry between right- and left-moving currents is indeed able to stabilize a chiral spin liquid phase.

It is interesting to note that Bauer *et al.* [70] proposed and found numerical evidence for a gapless chiral spin liquid in a model with staggered chirality on the kagome lattice that has the same symmetries as our model. The main difference is that their model contains three-spin interactions in the small (nearest-neighbor) triangles of the kagome lattice. In our approach of weakly coupled chains, the continuum limit of the three-spin interactions on small triangles only generates irrelevant couplings $\sim \mathbf{J}_1 \cdot (\mathbf{J}_2 \times \mathbf{J}_3)$ between chiral currents that belong to three different chains. For this reason, adding the staggered chirality in the triangles shown in Fig. 2 is a better starting point to approach the chiral spin liquid phase from weakly coupled crossed chains.

# 5 Perturbative renormalization group analysis

We derive perturbative RG equations [71] using the OPEs with the regularized delta function in the direction perpendicular to the chains, as exemplified by Eq. (25)-(28). The goal is to follow the RG flow out to length scales much larger than the lattice spacing. Importantly, increasing the intrachain short-distance cutoff also means increasing the number of chains a single chain crosses within that distance. Therefore, we must allow parallel chains to be correlated if they are separated by distances smaller than $\alpha \gg a$, where $\alpha$ is the short-distance cutoff used in the regularized delta function. Note that, since the correlations in the perpendicular direction are short-ranged, some of the coefficients that appear in the RG equations are nonuniversal [they depend in particular on the choice of the regularization $\Delta(x, \alpha)$]. But these coefficients are always of order one, and small changes do not affect the main features of our conclusions.

As a cutoff scheme, we integrate out short times $\alpha < v|\tau| < \alpha(1 + d\ell)$, but arbitrarily large distances, $-\infty < x, y < \infty$. The function $\Delta(x, \alpha)$ takes care of the convergence of all integrals over $x$ and $y$. We obtain the following set of RG equations.

$$\dot{\lambda}_1 = \lambda_1 + \frac{3}{8}\lambda_1(\lambda_3 + \lambda_4) + \frac{3}{16}c_1\lambda_2(\lambda_5 + \lambda_6) - \frac{3}{16}c_2\lambda_2(\lambda_7 + \lambda_8), \tag{48a}$$

$$\dot{\lambda}_2 = \lambda_2 - \frac{1}{8}\lambda_2(\lambda_3 + \lambda_4) + \frac{c_1}{8}\lambda_2(\lambda_5 + \lambda_6)$$
$$+ \frac{c_1}{16}\lambda_1(\lambda_5 + \lambda_6) - \frac{c_2}{16}(\lambda_1 + 2\lambda_2)(\lambda_7 + \lambda_8), \tag{48b}$$

$$\dot{\lambda}_3 = \frac{1}{4}(\lambda_3^2 + \lambda_4^2 + c_4\lambda_1^2 - c_4\lambda_2^2), \tag{48c}$$

$$\dot{\lambda}_4 = \frac{1}{2}\lambda_3\lambda_4, \tag{48d}$$

$$\dot{\lambda}_5 = \frac{c_1}{4}\lambda_5^2 + \frac{c_3}{16}\lambda_2^2 + \frac{c_5}{4}\lambda_1\lambda_2, \tag{48e}$$

$$\dot{\lambda}_6 = \frac{c_1}{4}\lambda_6^2 + \frac{c_3}{16}\lambda_2^2 + \frac{c_5}{4}\lambda_1\lambda_2, \tag{48f}$$

$$\dot{\lambda}_7 = \frac{c_2}{4}\lambda_7^2 + \frac{c_5}{16}\lambda_2^2, \tag{48g}$$

$$\dot{\lambda}_8 = \frac{c_2}{4}\lambda_8^2 + \frac{c_5}{16}\lambda_2^2, \tag{48h}$$

where $\dot{\lambda}_i = d\lambda_i/d\ell$ are the beta functions. The numerical coefficients $c_i$ are given in Appendix A. Note that the leading terms in the beta functions for $\lambda_1$ and $\lambda_2$ are first order in these parameters and have a positive eigenvalue, characteristic of relevant operators. For the other $\lambda_i$, the leading terms in the beta functions are quadratic, as expected for marginal operators [71].

We use Eqs. (47) and $\gamma_{\mathrm{bs}}^0 = -0.23$ [72] to set the initial values of the coupling constants. Without loss of generality, we take $J_\chi > 0$ (the opposite chirality, $J_\chi < 0$, is easily understood

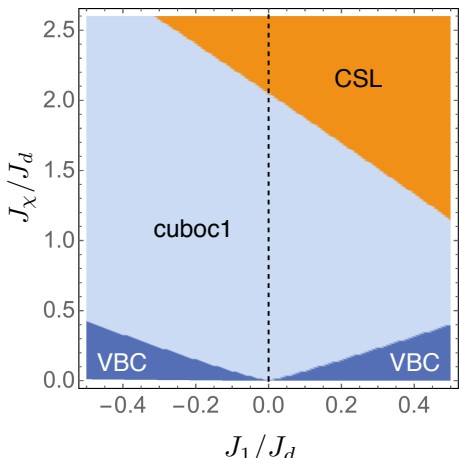

Figure 3: Ground state phase diagram obtained from the perturbative renormalization group approach. The CSL region corresponds to the gapless chiral spin liquid phase and VBC to the valence bond crystal.

by exchanging $R$ and $L$ in the discussion below). We solve the coupled RG equations numerically and stop the flow when the absolute value of one of the $\lambda_i$'s reaches 1. We find three possibilities as the parameters $J_1$ ($= J_2$) and $J_\chi$ are changed, leading to the phase diagram in Fig. 3. For small $J_\chi$, $\lambda_1$ reaches the strong coupling regime first; in this case, the system is in the spontaneously dimerized valence-bond-crystal phase discussed in Ref. [37]. As we increase $J_\chi$, there appears a transition to the *cuboc-1* phase governed by large $\lambda_2 > 0$. Finally, for larger $J_\chi$, the marginally relevant $\lambda_6$ (which is first order in $J_\chi$) is able to reach strong coupling before $\lambda_1$ and $\lambda_2$. Note that this phase appears at $J_\chi \gtrsim 2(J_d - J_1) \sim \mathcal{O}(J_d)$. The bare values of the dimensionless coupling constants are all smaller than unity in the entire parameter range shown in Fig. 3. However, they are close to the border of validity of the perturbative approach (for instance, $\lambda_6^0 \approx 0.52$ for $J_1 = 0.2 J_d$ and $J_\chi = 1.7 J_d$ inside the chiral spin liquid phase).

In Ref. [52], it was shown that a chiral spin liquid (in that case, the gapped Kalmeyer-Laughlin QSL) can be stabilized when a marginal operator that couples chiral currents in different chains reaches strong coupling before all competing relevant operators. Let us examine the effects of large $\lambda_6$ in our case. Recall that $\mathbf{J}_{qRe}$ describes the uniform mode between parallel chains, which, in the continuum limit, can be written as

$$\mathbf{J}_{qRe}(x, y) \sim \mathbf{J}_{qR}(x, y) + \mathbf{J}_{qR}(x, y - \alpha^*/2). \tag{49}$$

Here $\alpha^*$ is the scale at which $\lambda_6$ reaches strong coupling. The $\lambda_6$ interaction in Eq. 48f involves the operator

$$\mathcal{O}_q(x, y) \equiv \mathbf{J}_{qR}(-x, y) \cdot \mathbf{J}_{q+1, R}(x + y, x). \tag{50}$$

In the notation of abelian bosonization, the transverse part of the latter takes the form

$$\mathcal{O}_q^\perp(x, y) \sim \cos\left\{\sqrt{4\pi}\left[\varphi_{qR}(-x, y) - \varphi_{q+1, R}(x + y, x)\right]\right\}, \tag{51}$$

where $\varphi_{qR}(x, y)$ is the chiral boson that describes the right-moving (increasing $x$ of direction-$q$ chains) spin mode at transverse position $y$. The question then is what happens when $\lambda_6$ is the dominant term in the Hamiltonian. In comparison with the cosine perturbation in the standard sine-Gordon model, the operator in Eq. (51) looks more complicated because the field $\Phi_{qR}(x, y) \equiv \varphi_{qR}(-x, y) - \varphi_{q+1, R}(x + y, x)$ does not commute with itself at different positions. This raises the question whether the fields can be pinned consistently in a semiclassical analysis of the strong coupling limit $\lambda_6 \to \infty$. We note, however, that the chiral currents which

appear in the $\lambda_6$ interaction have integer scaling dimension. This implies that the operator in Eq. (51) *is local* in the sense that the commutator between the Hamiltonian density at different points vanishes for distances larger than the short-distance cutoff. Indeed, using the OPEs, we obtain

$$
\begin{aligned}
[\mathcal{O}_q(x,y), \mathcal{O}_{q'}(0,0)] = \lim_{\tau \to 0^+} &\left\{ \left[ \frac{\delta_{qq'}}{2} \frac{\delta^{ab}\Delta(y,\alpha^*)}{8\pi^2(\tau+ix)^2} + \frac{\delta_{qq'}}{2} \frac{i\epsilon^{abc}\sqrt{\Delta(y,\alpha^*)}}{2\pi(\tau+ix)} J_{qR}^c(0) \right. \right. \\
&+ :J_{qR}^a(x,y)J_{q'R}^b(0,0): \left] \times \left[ \frac{\delta_{qq'}}{2} \frac{\delta^{ab}\Delta(x-x',\alpha^*)}{8\pi^2[\tau-i(x+y)]^2} \right. \right. \\
&+ \frac{\delta_{qq'}}{2} \frac{i\epsilon^{abc}\sqrt{\Delta(x,\alpha^*)}}{2\pi[\tau-i(x+y)]} J_{q+1,R}^c(0) \\
&+ :J_{q+1,R}^a(x+y,x)J_{q'+1,R}^b(0,0): \left. \right] - (\tau \to -\tau) \right\}.
\end{aligned}
\tag{52}
$$

Taking the difference with $\tau \to -\tau$ terms and the limit $\tau \to 0^+$ introduces delta functions (or derivatives thereof) as usual in the Kac-Moody algebra [68]. It is then easy to verify that $[\mathcal{O}_q(x,y), \mathcal{O}_{q'}(0,0)] \approx 0$ for $|x|, |y| \gg \alpha^*$ due to the combination of delta functions and the regularized delta function $\Delta(x,\alpha)$.

Note that the short-distance cutoff $\alpha^*$ plays a crucial role as it sets the length scale below which the spins are correlated in all spatial directions. Above this scale, we can treat the local interactions in regions of size $\sim \alpha^*$ as independent from each other. This observation suggests that the right-moving bosons are locked in a uniform configuration that minimizes the energy associated with the cosine potential in Eq. (51).

The natural scenario is then to assume that the flow of $\lambda_6$ to strong coupling generates a gap for all right-moving bosonic modes in the system. To check the consistency of this assumption, we can show that the resulting theory is a *stable fixed point* of the renormalization group. If all $\varphi_{qR}$ are gapped out, we are left with gapless left-moving bosons $\varphi_{qL}$ at low energies. This is equivalent to a *chiral* $SU(2)_1$ WZW model in every chain. Importantly, recall that, even though the gapless modes are labeled as "left-movers", the actual directions of motion form 120 degree angles between different values of $q$ (which means, in particular, that there is no spin accumulation at a boundary of the system in an open geometry). The crucial point is that, without the $\varphi_{qR}$ modes, the relevant operators that involve $\mathbf{n}$ and $\varepsilon$ are forbidden in the low-energy theory. The absence of these perturbations stabilizes the chiral two-dimensional phase, in contrast with the fate of the time-reversal-symmetric sliding Luttinger liquid of spin chains [63]. We thus identify the corresponding phase (denoted by "CSL" in Fig. 3) as a gapless chiral spin liquid.

## 6 Properties of the gapless chiral spin liquid

### 6.1 Correlation functions

When the coupling constant $\lambda_6$ reaches strong coupling at a scale $\alpha = \alpha^*$, the spinon gap in the $R$ sector is of order $v/\alpha^*$. At distances larger than $\alpha^*$, all correlations that involve the gapped modes $\varphi_{qR}$ decay exponentially. This is the case for the fields $\mathbf{n}_q(x,y)$ and $\varepsilon_q(x,y)$.

The scale $\alpha^*$ also sets the finite range of correlations in the perpendicular direction for the gapless modes in the $L$ sector. Therefore, we obtain the spin-spin correlation

$$
\langle \mathbf{S}_q(j,l) \cdot \mathbf{S}_{q'}(j',l') \rangle \sim \langle \mathbf{J}_{qL}(x,y) \cdot \mathbf{J}_{q'L}(x',y') \rangle \sim -\frac{\Delta(y-y',\alpha^*)}{(x-x')^2} \delta_{qq'},
\tag{53}
$$

where we include the sign to emphasize that it is negative. Note the inverse-square-law decay along the chain directions. At sufficiently large distances, we can replace $\Delta(y,\alpha^*) \to \delta(y)$,

with the prescription $\delta(0) \to 1/\alpha^*$ if such a regularization is necessary. This form of the spin-spin correlation was also used in Ref. [62]. One can verify that, with this kind of scaling, residual current-current couplings at chain crossings are strictly irrelevant [62].

## 6.2 Low-energy density of states

The Fourier transform of the time-dependent local spin correlation yields

$$C(\omega) = \int_{-\infty}^{+\infty} dt\, e^{i\omega t} \langle \mathbf{S}_q(j,l,t) \cdot \mathbf{S}_q(j,l,0) \rangle \sim \int_{-\infty}^{+\infty} dt\, \frac{e^{i\omega t}}{(t-i\eta)^2} \sim \omega. \tag{54}$$

In a mean-field theory with noninteracting fermionic partons, this quantity maps onto the two-particle (or particle-hole) density of states [4,5]. The behavior $C(\omega) \sim \omega$ coincides with the expectation for a gapless QSL with a *spinon Fermi surface*, even though we have never assumed the existence of fermionic partons in our theory.

## 6.3 Entanglement entropy

Consider a finite subsystem $A$ with characteristic size $L$. The von Neumann entanglement entropy of subsystem $A$ with the rest of the system is given by $S = -\mathrm{tr}(\rho_A \ln \rho_A)$, where $\rho_A$ is the reduced density matrix of $A$. For $L \gg \alpha^*$, we can treat the system in the gapless chiral spin liquid phase as a chiral sliding Luttinger liquid, in which interchain couplings are irrelevant. For critical one-dimensional systems with periodic boundary conditions, it is well known that the entanglement entropy of the ground state scales with the subsystem size as [73,74]

$$S_{\mathrm{chain}} \sim \frac{c_L + c_R}{6} \ln(L/\alpha), \tag{55}$$

where $c_{R,L}$ are the chiral central charges of the conformal field theory and $\alpha$ is the short-distance cutoff. To estimate the entanglement of the two-dimensional system in our model, we can use the idea of Ref. [75] (adapted to one-dimensional modes in real space) and multiply the entanglement associated with a single chain by the number of chains that cross the boundary of $A$:

$$S \sim N_{\mathrm{chains}} \times S_{\mathrm{chain}} \sim \frac{c_L}{6} L \ln L, \tag{56}$$

where we used $N_{\mathrm{chains}} \sim L$ and the fact that only gapless left movers contribute to the logarithmic scaling of the entanglement entropy. Here, $c_L = 1$, corresponding to one free chiral boson in each chain, but the prefactor in Eq. (56) is actually nonuniversal. The behavior in Eq. (56) represents a logarithmic violation of the area law $S \sim L$ in two dimensions. Such violation is known to take place for gapless free fermion theories [76,77], but it also extends to strongly correlated systems described by a large number of one-dimensional modes [75].

## 6.4 Thermal Hall response

Magnetic systems with broken time-reversal and inversion symmetries can exhibit a nonzero thermal Hall conductivity $\kappa^{xy}$ [78]. In fact, a strong thermal Hall response has been proposed as a possible signature of fractionalization in QSLs [78]. However, we can show that our gapless chiral spin liquid has vanishing thermal Hall response as a result of the $2\pi/3$ rotational symmetry [43]. Within linear response theory, the thermal Hall conductivity at temperature $T$ is given by [79,80]

$$\kappa_{xy} = \kappa_{xy}^{\mathrm{Kubo}} + \frac{2M_E^z}{T}. \tag{57}$$

The first term is the contribution from the Kubo formula

$$\kappa_{xy}^{\text{Kubo}} = -\lim_{\omega \to 0} \frac{1}{\omega T} \text{Im} \Pi_{\text{ret}}^{xy}(\omega), \tag{58}$$

where

$$\Pi_{\text{ret}}^{xy}(\omega) = -i \sum_{\mathbf{r}} \int_0^\infty dt \, e^{i\omega t} \langle [J_E^x(\mathbf{r}, t), J_E^y(\mathbf{0}, 0)] \rangle \tag{59}$$

is the retarded correlation function mixing $x$ and $y$ components of the energy current density $\mathbf{J}_E(\mathbf{r})$. Here, $\mathbf{r}$ stands for conventional cartesian coordinates in the $(x, y)$ plane (as opposed to the coordinate system adopted in Fig. 1). The second term in Eq. (57) is a correction to the Kubo formula due to the so-called energy magnetization [80]. The latter is the solution to the differential equation

$$2M_E^z - T \frac{\partial M_E^z}{\partial T} = \frac{1}{2i} \left[ \frac{\partial}{\partial q_x} \langle\langle \mathscr{H}; J_E^y \rangle\rangle(\mathbf{q}) - \frac{\partial}{\partial q_y} \langle\langle \mathscr{H}; J_E^y \rangle\rangle(\mathbf{q}) \right] \tag{60}$$

where $\mathscr{H}(\mathbf{r})$ is the Hamiltonian density, $\langle\langle A; B \rangle\rangle(\mathbf{q}) \equiv \int_0^{1/T} d\tau \sum_{\mathbf{r}} e^{i\mathbf{q} \cdot \mathbf{r}} \langle A(\mathbf{r}, \tau) B(\mathbf{0}, 0) \rangle$ and $A(\mathbf{r}, \tau) = e^{H\tau} A(\mathbf{r}) e^{-H\tau}$ denotes imaginary time evolution. The solution to the differential equation must satisfy the boundary condition $\lim_{T \to 0} T \frac{\partial M_E^z}{\partial T} = 0$.

In the low-energy theory for the gapless chiral spin liquid, the energy current is carried by the gapless left movers. Using the expression for the energy current for spin chains [81], we can write

$$J_E^x(\mathbf{r}) = \frac{2\pi v}{3} \left[ \frac{1}{2} \mathbf{J}_{2L}^2(\mathbf{r}) + \frac{1}{2} \mathbf{J}_{3L}^2(\mathbf{r}) - \mathbf{J}_{1L}^2(\mathbf{r}) \right],$$

$$J_E^y(\mathbf{r}) = \frac{2\pi v}{3} \frac{\sqrt{3}}{2} \left[ \mathbf{J}_{3L}^2(\mathbf{r}) - \mathbf{J}_{2L}^2(\mathbf{r}) \right]. \tag{61}$$

Using the decoupling between the chains and the $R^2$ rotational symmetry, we have

$$\langle \mathbf{J}_{qL}^2(\mathbf{r}, t) \mathbf{J}_{q'L}^2(\mathbf{r}', 0) \rangle = \langle \mathbf{J}_{1L}^2(\mathbf{r}, t) \mathbf{J}_{1L}^2(\mathbf{r}', 0) \rangle \, \delta_{qq'}. \tag{62}$$

It then follows that $\kappa_{xy}^{\text{Kubo}}$ vanishes by symmetry. It is easy to verify that the energy magnetization correction also vanishes for the same reason.

## 6.5 Effect of weak disorder

The leading perturbation introduced by a static bond disorder potential is the coupling to the local energy density of the gapless modes:

$$H_{\text{dis}} = \sum_q \int dx dy \, W_q(x, y) \mathbf{J}_{qL}^2(x, y), \tag{63}$$

where $W_q(x, y)$ is treated as a Gaussian random variable,

$$\overline{W_q(x, y) W_{q'}(x', y')} = \mathscr{D} \, \delta(x - x') \delta(y - y') \delta_{qq'}. \tag{64}$$

Here, $\mathscr{D}$ represents the disorder strength. Following Ref. [82], we treat $H_{\text{dis}}$ using the replica trick. At leading order, the renormalization of the disorder strength can be determined by power counting in the replicated action

$$S_{\text{dis}} = -\mathscr{D} \int dx dy d\tau d\tau' \, \mathbf{J}_{qL}^2(x, y, \tau) \mathbf{J}_{qL}^2(x, y, \tau'). \tag{65}$$

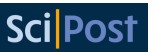

We find

$$\frac{d\mathscr{D}}{d\ell} = (4 - 2d_0)\mathscr{D} = -2\mathscr{D}\,, \tag{66}$$

where we used $d_0 = 3$ for the scaling dimension of the operator $\mathbf{J}_{qL}^2$ at the 2+1 dimensional fixed point. Therefore, weak disorder is strongly irrelevant, in striking contrast to the result for a single Heisenberg chain [83]. This property can also be obtained for sliding Luttinger liquids under certain conditions [69]. Here, the physical mechanism underlying the irrelevance of disorder is the absence of backscattering in the chiral spin liquid with gapped right movers.

## 7 Comparison with parton mean-field theory

So far, we have used coupled spin chains to construct a gapless chiral spin liquid whose properties turn out to be reminiscent of a spinon Fermi surface [84–87]. A natural question arises now: Is it possible to find a parton mean-field theory that recovers the physical properties of our state discussed in Section 6 ? Such a connection would be useful, for instance, in order to construct an approximate ground-state wave function for the gapless chiral spin liquid, and to evaluate its energy using variational Monte Carlo methods [5, 88].

### 7.1 Abrikosov fermion representation

We start this discussion by considering U(1) QSLs on the kagome lattice. Such wave functions have been studied quite extensively in efforts to determine the ground state of the nearest-neighbor Heisenberg antiferromagnet [44, 89–92]. The general idea is to fractionalize the spin operator into fermionic partons (so-called Abrikosov fermions) as [4, 5]

$$\mathbf{S}_j = \frac{1}{2} f_{j\alpha}^\dagger \boldsymbol{\sigma}_{\alpha\beta} f_{j\beta}\,, \tag{67}$$

where $f_{j\alpha}$, $\alpha = \uparrow, \downarrow$, are spin-1/2 annihilation operators, $\boldsymbol{\sigma} = (\sigma^1, \sigma^2, \sigma^3)$ denotes the Pauli matrices, and $j$ is a site index. This fractionalization leads to an emergent SU(2) gauge redundancy, and an enlargement of the Hilbert space. To recover physical states, one must impose the local single-occupancy constraint $\sum_\alpha n_{j\alpha} = 1$, where $n_{j\alpha} = f_{j\alpha}^\dagger f_{j\alpha}$. Although the original spin Hamiltonian is quartic in terms of the fermions, we can construct variational spin wave functions from the ground state $|\Psi_0\rangle$ of a quadratic trial Hamiltonian

$$H_{\mathrm{MF}} = \sum_{i,j,\alpha} t_{ij} f_{i\alpha}^\dagger f_{j\alpha}\,, \tag{68}$$

where $t_{ij} = t_{ji}^*$ are complex hopping amplitudes that specify the ansatz. A spin wave function describing a QSL (a resonating valence bond, or RVB state) is then obtained by $|\mathrm{QSL}\rangle = P_G|\Psi_0\rangle$, where $P_G = \prod_j (n_{j\uparrow} - n_{j\downarrow})^2$ is the Gutzwiller projector that imposes the local constraint.

We focus our analysis on ansätze that respect lattice translation and all symmetries discussed in Section 2. A natural starting point in the regime $J_d \gg |J_1| = |J_2|$ is a uniform hopping $t_{ij}$ restricted to the diagonals of the hexagons [36]. In this case, the parton theory describes free one-dimensional fermions. At the mean-field level, the spin-spin correlation is equivalent to the density correlation of the free fermions and behaves as $[A + B(-1)^r]/r^2$ at large distance. Going beyond mean field, one finds that the staggered part of the spin correlation for the Gutzwiller projected state decays as $(-1)^r/r$, in agreement with the exact result for the Heisenberg chain (up to logarithmic corrections). In fact, the projected wave function is the exact ground state of the Haldane-Shastry model [46, 47].

Turning on a hopping between first and second neighbors, we can obtain chiral spin liquids by threading nonzero U(1) gauge fluxes through plaquettes of the kagome lattice. The possible

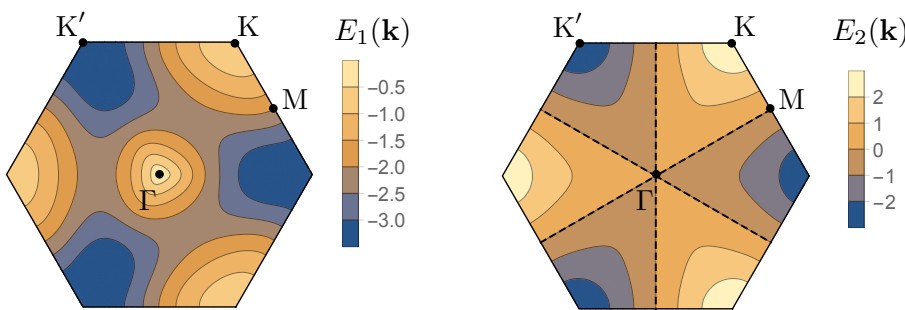

Figure 4: Contour plot of the spectra $E_n(\mathbf{k})$ in the QSL with Abrikosov fermions, No. 11 in Table IX of Ref. [45]. The used hopping parameters are $it_1 = t_2 = 0.7$, $t_d = i$. Symmetry relates the third band to the first one as $E_3(\mathbf{k}) = -E_1(-\mathbf{k})$. Fermi surfaces are indicated by dashed straight lines.

U(1) chiral spin liquids on the kagome lattice have been classified by a projective symmetry group analysis [36, 45]. To maintain the discussed symmetries, we focus on singlet states that break both time reversal $\Theta$ and lattice rotation $R$, but conserve $\Theta R$ and reflection $\sigma$ along the chain directions. These parton phases were listed as No. 9 – 14 in Table IX of Ref. [45]. This list includes states with various shapes of spinon Fermi surfaces. However, the main challenge in identifying a correct parton state is to recover the nonoscillating $1/r^2$ decay of spin correlations along chain directions in Eq. (53). For a circle-shaped spinon Fermi surface, the correlations decay as $\sim \cos(2k_F r + \phi)/r^3$ at the mean-field level, and this behavior is not affected by the Gutzwiller projection [84, 93]. In the case of a Dirac spectrum, the decay is even faster, $\sim 1/r^4$ [92].

Within the mean-field approximation, the spin-spin correlation in the parton state is given by

$$\langle \mathbf{S}_i \cdot \mathbf{S}_j \rangle = -\frac{3}{8} |G(\mathbf{r}_j - \mathbf{r}_i)|^2, \tag{69}$$

where $G(\mathbf{r}_i - \mathbf{r}_j) = \sum_\alpha \langle f_{i\alpha}^\dagger f_{j\alpha} \rangle$ is the parton Green's function. In the presence of a Fermi surface, it is dominated by

$$G(\mathbf{r}) \sim \int d^2k \, \Theta(\mu - \epsilon_k) e^{-i\mathbf{k}\cdot\mathbf{r}} \tag{70}$$

at large distances. Consider now a *straight segment* of Fermi surface (of length $2k_0$). In this case, the parton Green's function can be written as

$$G(\mathbf{r}) \sim \int^{k_F} dk_x \, e^{-ik_x r \cos\theta} \int_{-k_0}^{k_0} dk_y \, e^{-ik_y r \sin\theta}, \tag{71}$$

where $\theta$ is the angle between the space direction and the Fermi line normal. Upon integration, we find

$$G(r,\theta) \sim \frac{e^{-ik_F r \cos\theta}}{r \cos\theta} \frac{\sin(k_0 r \sin\theta)}{r \sin\theta}. \tag{72}$$

In the direction perpendicular to the Fermi line ($\theta \to 0$), the Green's function therefore displays the desired power law, $G(r) \sim 1/r$. Note that only a perfectly straight segment of Fermi surface results in this dominant contribution at large distance, and even a small curvature is expected to drastically change the power law.

Hence, we are looking for a parton ansatz with robust, symmetry-protected Fermi lines. Furthermore, the state should allow at least a hopping across the diagonals of the hexagons, and preferably also on first and second neighbors of the kagome lattice. Only state No. 11

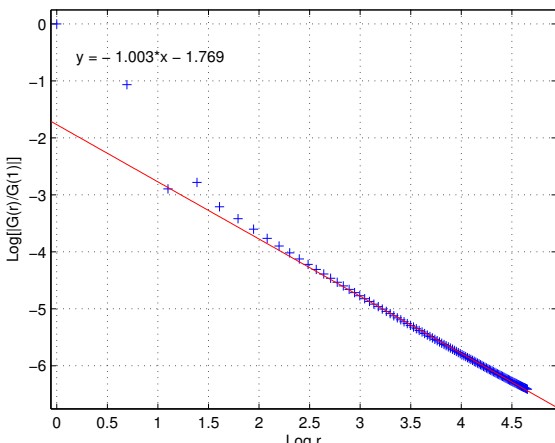

Figure 5: Log-plot of the Green's function $|G(r)/G(1)|$ along a chain direction in the complex parton state of Fig. 4. The spectrum with Fermi lines is shown there. The system is periodic with linear size $L = 512$. Correlations up to $r = 105$ are shown. The $1/r$-decay implies an algebraic spin-spin correlation $1/r^2$.

in Table IX of Ref. [45] fulfills all these requirements. As discussed in that paper, the lines in the Brillouin zone (BZ) left invariant by reflection $\sigma'_q$ are gapless, and they are protected by the projective symmetry of the parton ansatz. These Fermi lines run perpendicular to the chain directions, and the state therefore exhibits the correct $1/r^2$ spin-spin correlation. Away from these directions, the expected power law is $1/r^4$, or potentially $1/r^3$ in the presence of additional Fermi pockets.

The identified parton state has imaginary hopping on first neighbors and hexagon diagonals, and real hopping on second neighbors of the lattice. All hoppings change sign under $\pi/3$ rotation. It is a chiral spin liquid, since it has nontrivial U(1) fluxes $\pm\pi/2$ through first-neighbor triangles. The fluxes through second-neighbor triangles and hexagons vanish. The mean-field spectrum of this state has three bands with dispersion relations $E_n(\mathbf{k})$, $n = 1, 2, 3$. The spectrum for $it_1 = t_2 = 0.7$, $t_d = i$, is shown in Fig. 4, and the resulting Green's function along a chain direction in Fig. 5. For $|t_d| < 2|t_1|$, the bands $n = 1$ and $n = 3$ have gapless Dirac points at the center of the BZ. For $|t_d| > 2|t_1|$, additional circular Fermi pockets appear centered at K or K'. The $n = 2$ band has line Fermi surfaces along the $\Gamma-$M directions. The calculated spin-spin correlation function in the mean-field approximation displays the predicted power law $1/r^2$ at large distance along the chain direction (Fig. 5).

### 7.2 Majorana representation

As an alternative to the Abrikosov fermion fractionalization, spins can also be represented in terms of Majorana fermions [48, 94],

$$S_j^a = \frac{i}{2}\gamma_j^a\gamma_j^0. \tag{73}$$

The Majorana fermion operators obey $(\gamma_j^\mu)^\dagger = \gamma_j^\mu$ and $\{\gamma_j^\mu, \gamma_l^\nu\} = 2\delta^{\mu\nu}\delta_{jl}$, with $\mu, \nu \in \{0, 1, 2, 3\}$. In this representation, spin rotation acts as an SO(3) rotation of the vector $\boldsymbol{\gamma} = (\gamma^1, \gamma^2, \gamma^3)$, leaving $\gamma^0$ invariant. Spin is invariant under $\mathbb{Z}_2$ gauge transformations $\gamma_j^\mu \mapsto -\gamma_j^\mu$. The local constraint has the form

$$\gamma_j^1\gamma_j^2\gamma_j^3\gamma_j^0 = 1. \tag{74}$$

The general Hamiltonian for free Majorana fermions preserving spin-rotation symmetry is

$$H'_{\mathrm{MF}} = -i \sum_{ij} [u_{ij} \boldsymbol{\gamma}_i \cdot \boldsymbol{\gamma}_j + v_{ij} \gamma_i^0 \gamma_j^0], \tag{75}$$

where $u_{ij}$ and $v_{ij}$ are real parameters and obey $u_{ji} = -u_{ij}, v_{ji} = -v_{ij}$. Majorana fermions have purely imaginary mean fields, which implies that such QSLs necessarily break time reversal in the presence of loops with an odd number of sites. A variational wave function can be constructed by $|\mathrm{QSL}\rangle = P_D|\Psi'_0\rangle$, where $|\Psi'_0\rangle$ is the ground state of $H'_{\mathrm{MF}}$ and $P_D = \prod_j \frac{1}{2}(1 - \gamma_j^0 \gamma_j^1 \gamma_j^2 \gamma_j^3)$ is the projector to the physical subspace [48, 94].

Let us analyze a Majorana QSL on the kagome lattice with the symmetries discussed in Section 2. Let $u_1, u_2, u_d$ denote the mean fields between first neighbors, second neighbors, and along the diagonals of the lattice, respectively. The orientation of positive $u_{ij}$ determines the $Z_2$ gauge flux through triangles and hexagons. Symmetry under lattice reflection $\sigma_q$ (see Fig. 2) requires trivial flux through second-neighbor triangles. For Majorana fermions, this is only possible if $u_2 = 0$. This is in contrast to Abrikosov fermions, where (real) second-neighbor mean fields are allowed in certain projective symmetry groups.

We consider states in which the $Z_2$ flux changes sign between up- and down-pointing triangles. The resulting spectrum in the regime $u_d < 2u_1$ is similar to the result for Abrikosov fermions in Fig. 4. The gapless lines in the (middle) $n = 2$ band are protected by the reflection symmetry $\sigma_q$. Remarkably, Biswas *et al.* [43] encountered similar types of Fermi-surface lines in the single-band model of Majorana fermions on the triangular lattice. However, in the triangular lattice model, the spectrum exhibits a divergent single-particle density of states due to the cubic dispersion at the center of the BZ [43]. In our case, the density of states is finite and the two-particle density of states agrees with Eq. (54), except at the "critical" value $u_d = 2u_1$.

Within the Majorana parton mean-field theory, the spin correlation along the chain direction can be written as

$$\langle \mathbf{S}_q(j,l) \cdot \mathbf{S}_q(j+r,l) \rangle = \frac{3}{4} [\mathscr{G}(r)]^2, \tag{76}$$

where $r \in \mathbb{Z}$ in units of $a_\parallel$. Here we introduced the Majorana fermion Green's function

$$\mathscr{G}(r) = \langle \gamma_q(j,l) \gamma_q(j+r,l) \rangle, \tag{77}$$

where $\gamma_q(j,l)$ represents the Majorana fermion at position $(j,l)$ of sublattice $q$. The function $\mathscr{G}(r)$ is purely imaginary for $r \neq 0$. We confirm numerically that the slowest-decaying contribution to the Green's function behaves as $\mathscr{G}(r) \sim 1/r$. Therefore, the spin correlation is negative and decays as $1/r^2$ at large distances. In contrast to the U(1) chiral spin liquid discussed above, we do not need to require the presence of line Fermi surfaces to select the correct ansatz. Instead, for Majorana fermions, line Fermi surfaces appear as the only option once we impose the symmetry of the Hamiltonian.

# 8 Conclusion

We have studied a gapless chiral spin liquid phase in the extended Heisenberg model on the kagome lattice with an additional chiral interaction term $J_\chi$. Motivated by a spin model that had been proposed for the kapellasite material, we start from the limit where the strongest coupling is the exchange $J_d$ along the diagonals of the hexagons. In this limit, we obtain a model of weakly coupled crossed antiferromagnetic spin chains. We then analyze the effects of the interchain couplings using a perturbative renormalization group approach. This approach is based on an effective field theory for coarse-grained Heisenberg chains with short-range correlations in the transverse direction. We find that, for sufficiently strong chiral interaction,

the system can flow to a regime dominated by current-current couplings in one chiral sector. Assuming that this flow leads to a gap in this chiral sector, we can rule out perturbations driving valence-bond or magnetic order. As a result, the other chiral sector remains gapless and gives rise to a chiral sliding Luttinger liquid of spin. This coupled-chain construction is an alternative to the parton constructions and it does not rely on mean-field approximations to describe the quantum spin liquid. However, we showed that the properties of the gapless chiral spin liquid are consistent with a state in which the spins are fractionalized into (either Abrikosov or Majorana) fermions with Fermi surface lines that are protected by reflection symmetry.

We conjecture that the gapless chiral spin liquid phase described here extends beyond the regime where we can reliably use weakly coupled spin chains as a starting point in our analytical approach. At larger values of $J_\chi$, it is perhaps more appropriate to interpret the properties of the two-dimensional phase in terms of deconfined fermionic spinons with Fermi surfaces along straight lines in the BZ. It would be interesting to investigate the gapless chiral spin liquid in this regime using variational Monte Carlo or density matrix renormalization group methods. We believe that the chiral spin liquid identified in our work is the same phase as the one proposed by Bauer *et al.* for a different model of staggered chirality on the kagome lattice [70].

# Acknowledgements

We thank Simon Trebst, Oleg Starykh, Jesko Sirker, Hosho Katsura, Shoushu Gong, Claudio Chamon, Bela Bauer, and Dionys Baeriswyl for helpful discussions. RGP acknowledges support by CNPq.

# A Numerical coefficients

The numerical coefficients appearing in the bare coupling constants, Eqs. (44), are

$$
\begin{aligned}
C_1 &= \int_{1/2}^{1} d\tau \int_{-\infty}^{\infty} \frac{dx\, dy}{\pi} \left\{ \frac{\sqrt{\Delta(x,1/10)\Delta(y,1/10)}}{(\tau - ix)^2[\tau + i(x+y)]^2} + \frac{\sqrt{\Delta(x,1/10)\Delta(y,1/10)}}{(\tau - ix)^2[\tau - i(x+y)]^2} \right\} \\
&\approx 0.277,
\end{aligned}
\tag{78a}
$$

$$
\begin{aligned}
C_2 &= \int_{1/2}^{1} d\tau \int_{-\infty}^{\infty} \frac{dx\, dy}{\pi} \left\{ \frac{\sqrt{\Delta(x,1/10)\Delta(y,1/10)}}{(\tau - ix)^2[\tau + i(x+y)]^2} - \frac{\sqrt{\Delta(x,1/10)\Delta(y,1/10)}}{(\tau - ix)^2[\tau - i(x+y)]^2} \right\} \\
&\approx 0.007.
\end{aligned}
\tag{78b}
$$

The coefficients in the RG equations (48) are

$$c_1 = \int_{-\infty}^{+\infty} \frac{dx\,dy}{\pi} \frac{\sqrt{\Delta(x,1)\Delta(y,1)}}{(1-ix)[1+i(x+y)]} \approx 0.442 \,, \tag{79a}$$

$$c_2 = \int_{-\infty}^{+\infty} \frac{dx\,dy}{\pi} \frac{\sqrt{\Delta(x,1)\Delta(y,1)}}{(1-ix)[1-i(x+y)]} \approx 0.334 \,, \tag{79b}$$

$$c_3 = \int_{-\infty}^{+\infty} \frac{dx\,dy}{\pi} \frac{\sqrt{\Delta(x,1)\Delta(y,1)(1+x^2)[1+(x+y)^2]}}{(1-ix)[1+i(x+y)]} \approx 0.582 \,, \tag{79c}$$

$$c_4 = \int_{-\infty}^{+\infty} \frac{dx\,dy}{\pi} \sqrt{\Delta(x,1)\Delta(y,1)} \sqrt{\frac{1+(x+y)^2}{1+x^2}} \approx 0.701 \,, \tag{79d}$$

$$c_5 = \int_{-\infty}^{+\infty} \frac{dx\,dy}{\pi} \frac{\sqrt{\Delta(x,1)\Delta(y,1)(1+x^2)[1+(x+y)^2]}}{(1-ix)[1-i(x+y)]} \approx 0.383 \,. \tag{79e}$$

We emphasize that all these coefficients are nonuniversal as they depend on the parametrization of the short-range correlations and our choice of ultraviolet cutoff in the transverse direction.

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
