# Peer review of "Gapless chiral spin liquid from coupled chains on the kagome lattice"

_SciPost Physics, doi:SciPost Phys. 4, 004 (2018)_

## Round 1 · Referee Report · Tobias Meng (Referee 1) · 2017-6-16

Strengths

1- The authors approach the problem from different perspectives (coupled chains and parton mean fields), and find not inconsistent results. 2- The authors provide a detailed account of their calculation. 3- The topic is timely. 4- The approach is in principle a very good one to address questions such as "is it in principle possible to find this or that phase", as is the goal of the paper.

Weaknesses

1- There are technical concerns (see below) that have to be clarified. 2- The non-inconsistent results are not very specific (some out of many possible parton mean fields show the same behaviour as the coupled wires, it is left unclear if they are the appropriate ones).

Report

I have two main concerns.

The first one is related to one of the central results of the paper, namely the strong coupling analysis of the sine-Gordon term in Eq. (50). The authors state that this term leads, in its strong coupling phase, to the formation of a gapless chiral spin liquid, and the opening of a partial gap. Crucial for this conclusion is the pinning of the argument of the sine-Gordon term, see Eq. (51). However, what is the meaning of "right" and "left" for right-movers and left-movers in different q-chains, given that they are not parallel? This is obviously crucial for the relative sign of the commutator between fields of different qs, which the authors impose in the decoupled chain limit (where this question does not arise), and hence for whether the argument of the cosine can be pinned at all (which can be the case if the argument commutes with itself at different positions). Since this is not the most simple sine-Gordon situation, the authors have to clarify their technical treatment.

The second concern is about the comparison to the parton mean field construction. It is well known that, being a complicated mean field approach, parton constructions are somewhat arbitrary. They should hence be guided by some physical intuition: why is one parton construction chosen rather than another? In the precise case, the authors just pick from a list of mean field constructions the ones that have a spin correlation function decaying as 1/r^2. But is there any other reason for why the chosen patron states are good ones? In particular, I note that the authors find both a Z2 gauge field construction, and a U(1) gauge field construction to give the 1/r^2 decay, and hence two very different states. The conclusion can thus merely the hat one can find parton construction that happen to have the same decay of correlations as the coupled chain construction. Given the arbitrariness of parton mean fields, this is somewhat unsatisfactory. Can the authors find a more convincing argument as to why the parton mean fields they pick are relevant to their analysis?

Requested changes

1- The authors have to clarify their technical treatment of the sine-Gordon term for the entire paper to be considered scientifically correct.

2- The authors should expand their discussion of the parton mean fields, and be more precise about how much the comparison to parton mean fields is really telling us.

---

## Round 2 · Referee Report · Anonymous (Referee 2) · 2017-12-14

Strengths

1- The authors approach the problem from different perspectives (coupled chains and parton mean fields), and find not inconsistent results. 2- The authors provide a detailed account of their calculation. 3- The topic is timely. 4- The approach is in principle a very good one to address questions such as "is it in principle possible to find this or that phase", as is the goal of the paper.

Weaknesses

1- The bosonization procedure is still unclear

Report

The authors have responded well to the comments raised in the first round. The changes in the manuscript in my view did increase the understandability of the paper, and I thus recommend the work for publication.

I have one final comment on Eq. (50). The authors state that the argument of the cosine does not commute with itself at different positions. In bosonization, that means that the field cannot order - I hope the authors agree on that.

The authors then show that the cosine as a whole commutes with itself at different positions and state that "the most plausible scenario" is that the fields are gapped. It is mathematically unclear to me why this "most plausible scenario" is realised, given that the fields themselves cannot be pinned. Is there a way to understand this, or do I have to accept this as, really a guess? In other words, I find the physics plausible, but currently feel like there is a leap of faith from writing down Eq. (50) to saying that the right-moving fields are pinned. Sure, the phase where the right-movers are gapped might be a stable fixed point, but it feels to me like the authors are writing down a Hamiltonian they cannot solve, and then state that there is a stable fixed point, of which they do not show under which conditions the model realises it, and how that is related to the chiral interaction. Is that true?

I thus ask the authors to re-clarify this point.

Requested changes

1- Clear up the commutation of the argument of the cosine in Eq. (50).

  • validity: good
  • significance: good
  • originality: high
  • clarity: good
  • formatting: excellent
  • grammar: excellent

Author:  Rodrigo Pereira  on 2018-01-12  [id 200]

(in reply to Report 1 on 2017-12-14)
Category:
remark

We thank the referee for his/her helpful comments. We have added some clarification to emphasize that the full gap in one chiral sector is a conjecture since we cannot pin the bosonic fields in the usual way. Although the gap scenario seems plausible and physically compelling, this particular step in our analysis is indeed an assumption which we are not able to rigorously prove. Nonetheless, our derivation of the field theory from the lattice model and the perturbative renormalization group analysis are completely unbiased. Similarly, the calculation of physical properties is well controlled once we make this assumption about the strong coupling regime. It is worth mentioning that the lack of analytical techniques to handle non-commuting bosonized cosine terms is not unique to our problem. See, for instance, D. Bulmash et al., Phys. Rev. B 96, 045134 (2017) for a recent discussion of other examples.

We stress that the theory relying on our conjecture is consistent as it describes a stable fixed point of the renormalization group flow. Moreover, we clearly state that the chiral spin liquid phase can appear for strong chiral interaction, while valence-bond crystal and magnetic orders are expected to arise for small J_\chi. The validity of the conjecture could be tested by numerical simulations of the lattice model, by computing correlation functions that involve excitations of the gapped modes, such as dimer-dimer or the staggered magnetization mentioned in section 6.1.

---

## Round 2 · Author Response

The referee has raised very good points. First, we would like to clarify the meaning of right-moving bosons for crossed chains. We label R and L modes in the limit of decoupled chains in such a way that R (L) refers to the modes that propagate in the positive-x (negative-x) direction of q-chains as indicated by the arrows in Fig. 1. Thus, the direction of propagation of R/L modes at a given point depends on the q index. This coordinate system with three different positive-x directions may seem confusing, but it was introduced by Gong et al. in Ref. [37] because it is convenient for analyzing the threefold rotational symmetry of the Hamiltonian.

Second, we agree with the referee that our cosine potential differs significantly from a conventional sine-Gordon model. Nonetheless, we have rewritten the discussion around Eq. (50) to emphasize that the interactions are local in the sense that the operators at different positions (in two dimensions) commute when the distances are larger than the short-distance cutoff. (Note this does not happen for more general cosine potentials with non-integer scaling dimensions.) We interpret this as a sign that the right-moving bosons can be gapped out entirely, leaving only gapless left-moving bosons. We cannot prove the latter statement rigorously, but we show that the low-energy fixed point obtained within this assumption is stable and the theory is consistent. Clearly, the phase in which the coupling between chiral currents is the strongest interaction must be different from the magnetically ordered (cuboc) and valence-bond-crystal phases. In our opinion, our proposal of a gapless chiral spin liquid is the simplest and most plausible picture that emerges from the RG analysis.

Concerning the parton construction, we clarify that the point here was to show that there are at least two mean-field ansatze which are able to reproduce the properties of the chiral spin liquid derived from the coupled-chain approach. This is not a meaningless exercise because, once identified, the parton construction can be useful to analyze the properties of the chiral spin liquid in a regime where the starting point of weakly coupled chains is not reliable. The 1/r^2 decay of the correlation function indeed imposes a strong constraint on the theories that we can select, but it is not the only constraint. Our analysis is also guided by symmetry, as the mean-field ansatz must break time reversal and the reflection by lines perpendicular to the chain directions, but respect the symmetry of reflection by lines parallel to the chains. Out of large number of U(1) chiral spin liquids on the kagome lattice that have been classified in Ref. [45], we have narrowed the choice down to one. The other possible mean-field ansatz uses a Majorana fermion representation with a Z2 gauge structure. Selecting between the Z2 and U(1) parton constructions requires numerical methods to calculate the ground state energy (using e.g. variational Monte Carlo techniques) and it is beyond the scope of this paper. Most numerical studies of chiral spin liquids so far have focused on gapped phases (with the notable exception of Ref. [70]). We hope that our work will encourage the search for gapless chiral spin liquid phases.

---

## Round 2 · List of Changes

1. In the legend of Fig. 1, we now comment that the arrows indicate the positive-x directions, which become the directions of right-moving bosons for each q=1,2,3 in the continuum limit.

  2. To address the referee’s concerns about the relevant cosine potential, we have rewritten the discussion around Eq. (50).

  3. In section 6.4, we have included the energy magnetization correction in the expression for the thermal Hall conductivity, citing new references [79] and [80]. This correction does not change the conclusion that the thermal Hall response of the gapless chiral spin liquid vanishes by symmetry.

---

## Editorial Decision

published